# Effects of Garlic (*Allium sativum* L.) and Ramsons (*Allium ursinum* L.) on Lipid Oxidation and the Microbiological Quality, Physicochemical Properties and Sensory Attributes of Rabbit Meat Burgers

**DOI:** 10.3390/ani12151905

**Published:** 2022-07-26

**Authors:** Katarzyna Śmiecińska, Andrzej Gugołek, Dorota Kowalska

**Affiliations:** 1Department of Commodity Science and Processing of Animal Raw Materials, University of Warmia and Mazury in Olsztyn, Oczapowskiego 5, 10-719 Olsztyn, Poland; 2Department of Fur-Bearing Animal Breeding and Game Management, University of Warmia and Mazury in Olsztyn, Oczapowskiego 5, 10-719 Olsztyn, Poland; gugolek@uwm.edu.pl; 3Department of Small Livestock Breeding, National Research Institute of Animal Production, Balice, 32-083 Kraków, Poland; dorota.kowalska@iz.edu.pl

**Keywords:** rabbit meat burgers, shelf life, *Allium sativum* L., *Allium ursinum* L., lipid oxidation (TBARS), physicochemical properties, sensory attributes, microbiological quality

## Abstract

**Simple Summary:**

Modern consumers often look for convenience foods that are fast and easy to prepare, such as burgers. Burgers can be made from different types of meat with various additives. Due to its high nutritional value and low calorie and fat content, rabbit meat can be used as a raw material for preparing burgers. Consumers are becoming increasingly aware of their food choices, and they tend to avoid synthetic and artificial ingredients in meat products, which can be toxic. In order to meet customer expectations, producers often replace synthetic additives with natural alternatives that extend the shelf life of meat products and enhance their taste and aroma. Garlic is widely used in the meat processing industry, and it pairs well with rabbit meat. The aim of this study was to evaluate the quality of rabbit meat burgers with the addition of garlic powder, ramsons powder or their combination. It was found that garlic powder and ramsons powder can be added to rabbit meat burgers to extend their shelf life and improve their sensory attributes.

**Abstract:**

The aim of this study was to evaluate the quality of rabbit meat burgers with the addition of garlic (*Allium Sativum* L.) powder (G), ramsons (*Allium ursinum* L.) powder (R) or their combination (GR). The effects of additives on lipid oxidation, color parameters, microbiological quality and organoleptic properties of raw and oven-baked burgers were analyzed before and after refrigerated storage. Four meat formulations were prepared: control (C)—without additives, with the addition of G (0.35 g/100 g of meat), R (0.35 g/100 g of meat) and GR (0.35 g/100 g of meat each). The addition of GR induced an increase in pH and TBARS values in raw and oven-baked burgers. The pH of raw and oven-baked burgers was also affected by storage time (ST), and it was lower after 7 days of storage (ST7) than before storage (ST0). TBARS values were higher at ST7 only in raw burgers. The addition of R and GR decreased the values of color parameter L* (lightness) relative to G and C in raw and oven-baked burgers. The greatest changes in parameter a* (redness) were observed after the addition of R and GR, both before and after heat treatment. The values of parameter b* (yellowness) increased after the addition of R, GR (raw and oven-baked burgers) and G (raw burgers). In raw burgers, color saturation (C*) was higher in groups R and GR than in groups C and G, and the value of hue angle (h°) was lower in burgers with GR than in those with G and R. In oven-baked burgers, the values of C* and h° were lower in group GR than in the remaining treatments (C, G and R). In raw burgers, ST had no effect on the values of L*, whereas the values of parameters a*, b*, C* and h° were lower at ST7 than at ST0. In oven-baked burgers, the values of L* were higher at ST0 than at ST7, and the values of a*, b*, C* and h° were higher at ST7 than at ST0. The tested additives had no influence on the presence of off-odors in raw burgers. This parameter was affected by ST, and its value was lower at ST0 than at ST7. The appearance and overall acceptability of burgers were affected only by additives, and raw burgers containing GR received the lowest scores. After heat treatment, control burgers scored lowest for all attributes, whereas burgers with the addition of R and GR received the highest scores. The analyzed additives had no effect on the growth of *Enterobacteriacea,*
*Pseudomonas* spp., lactic acid bacteria or total aerobic psychrotrophic bacteria. However, the counts of all identified bacteria increased at ST7. In conclusion, garlic powder and ramsons powder can be added to rabbit meat burgers to extend their shelf life and improve their eating quality.

## 1. Introduction

The diverse consumer needs and growing consumer expectations regarding high product quality and food safety play a very important role on the contemporary food market. There is a growing demand for food products that can be easily and quickly prepared under different conditions. Convenience foods meet these requirements. Products containing different types of ground meat with various additives, such as burgers, constitute a large category of convenience foods. Rabbit meat is an excellent raw material for the production of burgers on account of its high nutritional value and low calorie and fat content.

Rabbit meat is rich in protein with high biological value, and it contains less fat, saturated fatty acids and cholesterol than other types of meat [1,2]. For this reason, rabbit meat has been recommended by the World Health Organization (WHO) as an ideal food for children [3]. In retail, rabbit meat is usually available as whole rabbit carcasses or selected cuts, such as hind legs and loin [4,5]. However, rabbit carcasses and cuts are not highly popular, especially among children (under 18), and 40.9% of young consumers do not like this type of meat, whereas 30.9% never eat rabbit meat at home [6]. The consumption of rabbit meat could be increased by introducing processed products, such as burgers, nuggets or sausages, on the retail market [7,8]. Various additives can be applied to enhance the sensory properties of rabbit meat products [5], and natural antioxidants can be used [9,10,11] to extend the shelf life of these products [6].

Antioxidants are applied in the meat processing industry to prevent lipid oxidation which can lead to undesirable changes in quality attributes (such as color, taste and aroma) as well as food spoilage [12]. Synthetic and natural antioxidants can also deliver health benefits for consumers [13,14,15]. Fruits, vegetables, cereals, spices, herbs and tea are natural sources of antioxidants [10,16,17,18]. Consumers are increasingly likely to make informed food choices, and they avoid meat products containing potentially toxic synthetic additives [19,20]. Many food producers replace synthetic additives with natural substances to meet customer expectations [16]. Natural additives offer an attractive alternative to synthetic compounds because they contain antioxidants that prolong shelf life, and they also enhance the taste and aroma of meat products [21,22,23].

The genus *Allium* contains 450 species, including garlic (*Allium sativum* L.) and ramsons (*Allium ursinum* L.) [24]. *Allium* species are abundant in essential amino acids, vitamins (C, E, B_1_, B_2_, B_6_), minerals (P, Zn, Se, K, Fe, Mg, Ca, Na) and sulfur-containing compounds [25,26,27] with antimicrobial activity, and they are often regarded as natural antibiotics [14]. Due to its exceptional composition, as well as anti-lipidemic, anti-thrombotic, anti-hypertensive, anti-atherogenic and anti-glycemic effects [28], garlic had been recognized as an effective remedy for the treatment of many diseases already in ancient Egypt. Garlic has been used in traditional Chinese medicine for more than 3000 years, and its popularity has not diminished in the modern world [24]. Garlic is also applied in the meat processing industry as an additive in the production of cold cuts and ready-made foods [13]. Garlic contains biologically active compounds such as sapogenins, saponins and flavonoids which remain stable during high-temperature processing and storage [29]. Flavonoids such as flavones and quercetins, as well as sulfur-containing compounds such as allyl-cysteine, diallyl sulfide and allyl trisulfide, have antioxidant properties [30]. Garlic and ramsons also possess antimicrobial properties, and they are used in the food processing industry to stabilize foods and enhance their sensory attributes [24,31]. Garlic (*Allium sativum* L.) is valued for its unique aroma, flavor and pungency [24], and it widely renowned for its ability to stimulate digestion [15]. However, not all consumers enjoy the taste and aroma of garlic. Ramsons (*Allium ursinum* L.) has a less distinctive flavor and aroma [14], and it provides a good substitute for garlic (*Allium sativum* L.). *Allium ursinum* is also known as bear’s garlic, and this name is derived from the Latin word *ursus*, meaning bear. According to folk tales, bears feast on ramsons to purge toxins and regain strength after a long winter’s sleep [32]. *Allium ursinum* L. is a ubiquitous species in Europe, Asia Minor, Caucasus and Siberia, all the way to the Kamchatka Peninsula [33].

Considerable research has been conducted on the use of natural additives to improve the physicochemical and sensory attributes of rabbit meat products and extend their shelf life [8,9,34,35]. However, the use of ramsons (*Allium ursinum* L.) as an additive in rabbit meat products has never been studied, and products enhanced with ramsons and garlic (*Allium sativum* L.) have never been compared in the literature. In the present study, both plant species were added to rabbit meat burgers because, apart from their numerous advantages, garlic and ramsons go well with rabbit meat [8].

The aim of this study was to evaluate the quality of rabbit meat burgers with the addition of garlic (*Allium Sativum* L.) powder (G), ramsons (*Allium ursinum* L.) powder (R) or their combination (GR). The effects of additives on the lipid oxidation, color parameters, microbiological quality and organoleptic properties of raw and oven-baked burgers were analyzed before and after refrigerated storage.

## 2. Materials and Methods

### 2.1. Materials

Experimental material was sampled from 40 rabbit carcasses purchased in the abattoir on a large rabbit farm in southern Poland. The weight of a cold carcass without a head and giblets was 1198.15 ± 31.86 g (means ± SEM). New Zealand White rabbits were raised under standard conditions, and were fed ad libitum pelleted diets formulated to meet the nutrient requirements of growing broiler rabbits [36], which contained 15.5% total protein, 4.1% crude fat and 17.1% crude fiber. Metabolic energy (ME) of feed was 10.2 MJ.

At 90 days of age, when their body weight was 2503.77 ± 22.48 g (means ± SEM), the animals were fasted for 24 h, and were sacrificed according to the standard guidelines for euthanizing experimental animals (rabbits were stunned and bled, and the entire procedure took approx. 2 min.) [37]. Slaughtered rabbits were skinned and eviscerated, and then they were chilled in a refrigerating chamber at a temperature of 2 °C for 24 h. Chilled carcasses were dressed. The head was dissected along the occipital joint; the forepart was dissected between the 7th and 8th thoracic vertebrae; and the loin was dissected between the 6th and 7th lumbar vertebrae. The hind part, including the perisacral area and hind legs (HL), was the remaining part of the dissected carcass [38]. Hind legs were removed from the hind part, deboned, vacuum-packaged (99% vacuum is equal to 1.3 KPa) in bags (polyamide/polyethylene film) with enhanced gas barrier performance (permeability: O_2_ = 25 cm^3^/m^2^/24 h/0.1 Mpa; CO_2_ = 85 cm^3^/m^2^/24 h/0.1 Mpa; N_2_ = 7 cm^3^/m^2^/24 h/0.1 Mpa; moisture vapor < 3 g/m^2^/24 h) with the use of the PP15 (MGO) Tepro Vacu Tronic 2000 vacuum packaging machine (Tepro S.A., Koszalin, Poland), frozen at −22 °C (slow freezing, 0.2–1 cm/h for 20 h), and stored at this temperature for 21 days. Then HL were thawed in a refrigerating chamber (in the atmospheric air) at a temperature of 2 °C and relative air humidity of 85% for 20 h. After thawing, HL were randomly divided into four portions which were ground in a meat grinder to pass through a 2.7 mm mesh screen, and thoroughly mixed. The first, control portion (C), contained no additives; food additives approved for use and available in retail outlets were added to the remaining portions: garlic (*Allium sativum* L.) powder (0.35 g/100 g of meat) (G), ramsons (*Allium ursinum* L.) powder (0.35 g/100 g of meat) (R) and both garlic powder and ramsons powder (0.35 g/100 g of meat each) (GR). The amounts of additives were determined based on the results of a preliminary study and a sensory analysis performed by university staff members. Meat was thoroughly mixed by hand with the tested additives, and burgers were formed with the use of a commercial manual burger press (BIOWIN 311411, BROWIN*,* Łódź*,* Poland). Each burger had a weight of 100 g and a diameter of 11 cm. Each of the four meat portions was formed into 40 burgers, 160 burgers in total. Burgers were placed on Styrofoam trays and wrapped in polyethylene foil. Some of the burgers were evaluated before storage (ST0) and others were evaluated after 7 days of storage (ST7) at a temperature of 4 °C. In each batch, raw burgers (10 burgers at ST0 and 10 burgers at ST7) and oven-baked burgers (10 burgers at ST0 and 10 burgers at ST7) were assessed. Burgers were baked in a pre-heated electric oven (Amica Wronki S.A., model 1143.4TdYDbHaOQVS, Wronki, Poland), at a temperature of 163 °C until their internal temperature reached 71 °C (Electrolux E4KTD001 digital cooking thermometer, Stockholm, Poland); during baking, burgers were rotated every 4 min. Raw and oven-baked burgers (ST0 and ST7) were analyzed to determine their pH, lipid oxidation (TBARS), color parameters and sensory properties, and raw burgers were analyzed to determine cooking loss and microbiological quality.

### 2.2. Methods

#### 2.2.1. pH

The pH values of rabbit meat burgers were measured using a Polilyte Lab combination electrode (Hamilton Bonaduz AG, Bonaduz, Switzerland) and a 340i pH-meter equipped with a TFK 325 temperature sensor (WTW Wissenschaftlich-Technische Werkstätten, Weilheim, Germany).

#### 2.2.2. Lipid Oxidation (TBARS)

The rate of lipid oxidation was determined in a thiobarbituric-acid-reactive substances (TBARS) assay [39]. Absorbance was measured with the Specord^®^ 40 spectrophotometer (Analytik Jena AG, Jena, Germany) at a wavelength of 532 nm. The TBARS value was expressed as mg of malondialdehyde (MDA) per kg of meat.

#### 2.2.3. Cooking Loss

After roasting in an electric oven (163 °C until internal temperature of 71 °C was reached, rotation every 4 min.), burgers were stored at room temperature for several minutes, and their surface was gently dried with paper. Cooking loss (%) was calculated as follows: cooking loss = [(w_b_ − w_a_)/w_b_] × 100, where w_b_ and w_a_ denote burger weight before and after heat treatment, respectively [40].

#### 2.2.4. Color

Color was determined based on the values of CIELAB coordinates, L* (lightness), a* (redness), b* (yellowness), C* (chroma) and h° (hue) [41]. The color space parameters L*, a* and b* were measured by the reflectance method using a HunterLab MiniScan XE Plus spectrocolorimeter (Hunter Associates Laboratory Inc., Reston, VA, USA) with standard illuminant D65, a 10 standard observer angle and a 2.54 cm-diameter aperture. Each data point was the mean of three replications measured on the surface of the burgers at randomly selected locations. The apparatus was standardized using black and white standard plates. The values of C* were calculated from the following formula: C* = (a*^2^ + b*^2^)^1/2^. The values of h° were calculated from the following formula: h° = tan^−1^(b / a).

#### 2.2.5. Sensory Analysis

The sensory properties of raw and oven-baked burgers were evaluated by six trained panelists selected for their sensory sensitivity [42]. Prior to sample evaluation, the panelists participated in orientation sessions to familiarize themselves with the sensory properties of raw and oven-baked burgers. The panelists assessed samples in individual compartments. Fluorescent white lights (500 lx) that simulated daylight, installed at a height of approximately 1 m, were used to evenly illuminate the table. Relative minimum air humidity of 60% and temperature of 21 °C were maintained in the panel room. Rabbit meat burgers were baked in an electric oven, at a temperature of 163 °C until internal temperature reached 71 °C; during baking, burgers were rotated every 4 min. [40]. Burgers were divided into six equal parts and they were presented to the panelists immediately after heat treatment. All sensory attributes of encoded samples were evaluated during a single session. The panelists were offered still mineral water with a neutral taste and aroma for cleansing the palate. Burgers were rated on a 10 cm long unscaled line with the sensory properties of burgers. A total of seven parameters were assessed: appearance, aroma intensity (defined as the intensity of the characteristic aroma of rabbit meat burgers), flavor intensity (defined as the intensity of the characteristic flavor of rabbit meat burgers), hardness, juiciness, off-odors, off-flavors and the overall acceptability of samples (the latter parameter was assessed on a 9-point structured scale: 1—extremely negative, 5—neither negative nor positive, 9—extremely positive). Raw burgers were analyzed for their appearance, the presence of off-odors and overall acceptability [8].

#### 2.2.6. Microbial Analysis

##### Sample Preparation

For the preparation of samples, 10 g minced meat was aseptically weighed and transferred to 90 mL sterile saline (0.85% NaCl) and homogenized with a stomacher (Masticator Homogenizator Silver, IUL S.A., Barcelona, Spain). Homogenized samples were serially diluted using the same diluent [1:10 (*v*/*v*)]. For microbial enumeration, 1 mL of each dilution was transferred to a Petri dish and a sterile medium was poured on top.

Enumeration of *Pseudomonas* sp., mesophilic lactic acid bacteria, psychrotrophic bacteria and rods of the family *Enterobacteriaceae*.

The pour-plating procedure was used for the enumeration of *Pseudomonas*, mesophilic lactic acid bacteria, psychrotrophic bacteria and rods of the family Enterobacteriaceae. Cetrimide LAB-AGAR^TM^ (BioMaxima SA, Lublin, Poland) and VRBD agar (Merck, Darmstadt, Germany) were incubated at 37 °C ± 1 °C for 24 h ± 1 h, respectively, for the enumeration of Pseudomonas sp. and Enterobacteriaceae. Plate count agars for the enumeration of psychrotrophic bacteria were incubated at 21 °C ± 1 °C for 25 h ± 1 h. Mesophilic lactic acid bacteria were enumerated using the pour-plating procedure; MRS agar (Merck, Darmstadt, Germany) was incubated at 37 °C ± 1 °C for 72 h ± 3 h in anerobic conditions.

##### Identification of Bacterial Colonies

Within each bacterial group, the most common colonies were identified by matrix-assisted laser desorption and ionization (MALDI). Measurements were performed using a VITEK^®^ MS (bioMérieux, Marcy l’Etoile, France) with an acceleration voltage of 200 kV, mass range of 2–20 kDa, laser frequency of 50 Hz and extraction delay time of 200 ns. All mass fingerprints were analyzed with the VITEK^®^ MS v2.0 MALDI-TOF mass spectrometry system and the V2.0 research use only (RUO; SARAMIS version 4.13) database (bioMérieux, Marcy l’Etoile, France). Isolates were tested in duplicate using the direct transfer protocol according to the manufacturers’ recommendations. Briefly, the isolates were cultured for 48 h at 30 °C on TSA (Merck, Darmstadt, Germany). Next, a small amount of a colony was transferred to a metallic MALDI plate and covered with 1 µL of CHCA matrix (α-Cyano-4-hydroxycinnamic acid) (bioMérieux, Marcy l’Etoile, France). After crystallization of the matrix solution, the target was loaded into the MALDI-TOF MS chamber, and the analysis was started.

### 2.3. Statistical Analysis

The results were processed statistically using the STATISTICA program, version 13.3 (TIBCO Software Inc., Palo Alto, CA, USA). The normality of data distribution was checked by the Shapiro–Wilk test. The effects of additives, storage time and their interaction on the analyzed parameters of rabbit meat burgers were determined by two-way analysis of variance (ANOVA). If interactions between the experimental factors were not found (*p* > 0.05), the significance of differences between group means was estimated by Tukey’s test. When an interaction was noted (*p* ≤ 0.05), one-way ANOVA was performed, and group means were compared by Tukey’s test. The significance of differences between group means was determined at *p* ≤ 0.05. Variability was expressed as the standard error of the mean (SEM).

## 3. Results

### 3.1. pH, TBARS Values and Cooking Loss of Raw and Oven-Baked Burgers

An analysis of the effects exerted by different additives (A): G (garlic), R (ramsons) and GR (garlic and ramsons) on the pH of burgers revealed that GR caused an increase in its values in both raw and oven-baked burgers (Table 1). Raw burgers with the addition of GR had higher (*p* < 0.001) pH values than control burgers without additives (C), and oven-baked burgers with the addition of GR had higher (*p* < 0.001) pH values than burgers in the other groups (C, G and R). Storage time also affected pH values measured in raw and oven-baked burgers, which were lower (*p* < 0.001) after storage (ST7) than before storage (ST0).

TBARS values in raw and oven-baked burgers with the addition of GR were higher (*p* < 0.001) than in the remaining groups (C, G and R) (Table 1). The addition of R increased (*p* < 0.001) TBARS values relative to group C in raw burgers, and relative to groups C and G in oven-baked burgers. These findings indicate that garlic provided more effective antioxidant protection than ramsons. Storage time contributed to an increase in TBARS values only in raw burgers; TBARS values were higher at ST7 (*p* < 0.001). Neither the additives nor ST exerted a significant (*p* > 0.05) effect on cooking loss (*p* = 0.084 and *p* = 0.650, respectively) (Table 1).

Interactions between additives and ST were not significant (*p* > 0.05) for the parameters presented in Table 1.

### 3.2. Color Parameters of Raw and Oven-Baked Burgers

The tested additives considerably affected the color of raw and oven-baked burgers (Table 2). The addition of R and GR decreased (*p* < 0.001) the values of color parameter L* relative to G and C in raw and oven-baked burgers. In raw burgers, all additives decreased (*p* < 0.001) the contribution of redness (a*), relative to group C. In oven-baked burgers, the values of parameter a* were lower (*p* < 0.001) in groups R and GR than in groups C and G. The greatest changes in parameter a* were observed after the addition of R and GR, both before and after heat treatment. The values of parameter b* (yellowness) increased (*p* < 0.001) after the addition of R, GR (raw and oven-baked burgers) and G (raw burgers), compared with control burgers. The differences in the contribution of color components a* and b* led to differences in the average values of color saturation (C*) and hue angle (h°) (Table 2). In raw burgers, the values of C* were higher (*p* < 0.001) in groups R and GR than in groups C and G, and the value of h° was lower (*p* = 0.001) in burgers with GR than in those with G and R. In oven-baked burgers, the values of C* and h° were lower (*p* < 0.001) in group GR than in the remaining treatments (C, G and R). In raw burgers, ST had no effect (*p* > 0.05) on the values of L*, whereas the values of parameters a* (*p* = 0.023), b* (*p* < 0.001), C* (*p* < 0.001) and h° (*p* = 0.010) were lower at ST7 than at ST0. In oven-baked burgers, the values of L* were higher (*p* = 0.037) at ST0 than at ST7, and the values of a*, b*, C* and h° were higher at ST7 than at ST0 (*p* < 0.001 for a*, b* and C*, *p* = 0.002 for h°).

In raw burgers, an interaction (*p* = 0.011) between additives and ST was found for the value of a*, which was higher (*p* < 0.001) in the control group before storage than in burgers with the addition of GR after storage.

### 3.3. Sensory Properties of Raw and Oven-Baked Burgers

An analysis of the sensory properties of raw burgers revealed that the tested additives (G, R and GR) had no influence (*p* > 0.05) on the presence of off-odors. This parameter was affected by ST, and its value was lower (*p* = 0.010) at ST0 than at ST7. However, despite differences, the noted values were very low in both groups. The appearance and overall acceptability of burgers were affected only by additives (*p* < 0.001), and raw burgers containing GR received the lowest scores. After heat treatment, control burgers scored lowest for all attributes, whereas burgers with the addition of R and GR received the highest scores.

These results indicate that a combination of G and R had a more beneficial influence on the sensory properties of oven-baked burgers than G alone. The addition of R, compared with G, had a more positive effect (*p* < 0.001) on the appearance, hardness, juiciness and overall acceptability of burgers. Despite significant differences between off-odors and off-flavors identified in oven-baked burgers with various additives (*p* = 0.040 and *p* < 0.001, respectively), they were detected at very low levels.

Both parameters were also affected by ST. The concentrations of off-odors and off-flavors increased in burgers stored for 7 days (*p* < 0.001 and *p* = 0.001, respectively), but the noted values were very low. Moreover, ST affected the appearance and overall acceptability of oven-baked burgers. The values of these parameters were higher (*p* < 0.001 and *p* = 0.002) at ST7 than at ST0. The values of flavor intensity, hardness and juiciness (*p* < 0.001) were higher before than after storage. The aroma intensity of oven-baked burgers was not affected by ST (*p* > 0.05).

Interactions between additives and ST were not significant (*p* > 0.05) for the parameters presented in Table 3.

### 3.4. Microbiological Quality of Raw Burgers

The analyzed additives had no effect on the growth of *Enterobacteriacea* (*p* = 0.796)*,*
*Pseudomonas* spp. (*p* = 0.840), lactic acid bacteria (*p* = 0.388) or total aerobic psychrotrophic bacteria (*p* = 0.900). However, the counts of all identified bacteria increased at ST7 (*p* < 0.001).

Interactions between additives and ST were not significant (*p* > 0.05) for the parameters presented in Table 4.

## 4. Discussion

### 4.1. Physicochemical Properties and Lipid Oxidation (TBARS)

Apart from bacterial growth, lipid oxidation is one of the major processes responsible for quality deterioration in meat and meat products during storage, processing and handling, which ultimately shortens their shelf life [43,44,45]. Lipid oxidation can affect the flavor, color, consistency and nutritional value of meat [21,22]. Undesirable changes in the quality and acceptability of meat and meat products result from the decomposition of secondary oxidation products such as short-chain aldehydes, ketones and malonaldehyde, which can be harmful to human health [22,43,46]. In stored meat, lipid oxidation contributes to an increase in TBARS values, total bacterial counts and metmyoglobin concentration, which is usually accompanied by a decrease in pH levels and the values of color lightness (L*) and redness (a*) [47]. Some of these processes were also observed in this study, where the pH of raw and oven-baked burgers decreased after storage. Mancini et al. [35] noted a slight increase (*p* = 0.002) in the pH of raw rabbit burgers stored for 7 days, which could result from an increase in ammoniacal nitrogen levels, and protein and amino acid degradation by Gram-negative bacteria.

In the present study, TBARS values increased only in raw burgers, and lipid autooxidation in raw and oven-baked burgers was most effectively inhibited by garlic powder. The antioxidant activity of ramsons powder was comparable with that of garlic powder only in raw burgers. It should be stressed, however, that average TBARS values in experimental groups as well as in individual samples did not exceed 5 mg MDA, which is considered as the limiting threshold for acceptability of oxidation in meat [48]. An increase in TBARS values during chilled storage was also observed in other studies investigating lamb burgers [47], deer burgers [49], rabbit burgers [8], pork burgers [50] and beef patties [51], pointing to the instability of ground meat products, including those packaged under a modified atmosphere (MAP).

The high antioxidant potential of natural food additives [9,10,11,21,49,52], including garlic [24,53,54,55,56] and ramsons [31,32,55], has been well documented. However, the addition of ramsons powder to meat stuffing has not been investigated to date, and only a few studies have evaluated the quality of meat products containing garlic [8,57,58]. Previous research has shown that the antioxidant capacity of garlic is determined by its form (degree of processing), concentration and the type of food product. Sallam et al. [58] demonstrated that fresh garlic, garlic powder and garlic oil exerted antioxidant and antimicrobial effects on raw chicken sausages stored at 3 °C. Mancini et al. [8] found that garlic powder added to rabbit burgers at 0.25% provided partial antioxidant protection, but further research is needed to elucidate whether different garlic products or concentrations could deliver more beneficial effects.

In the current study, neither the tested additives nor storage time affected cooking loss. The high water-holding capacity of burgers can be attributed to their relatively high pH values and the grinding process, which can also improve meat’s ability to retain water [59]. Heat treatment, including its type and parameters (temperature, time, heating medium), is very important for maintaining the high quality of meat products because it can affect their chemical, physical, biological and sensory properties, and consumer acceptability [60,61]. The rate of lipid oxidation in meat is also affected by the type and parameters of heat treatment [61,62]. Cooking, which takes place at a relatively low temperature (approx. 100 °C), usually induces minor changes in meat lipids. In turn, baking and frying involve high temperatures and may adversely affect lipid composition [63]. The antioxidant potential of food ingredients can decrease, increase or remain unchanged during technological processes. New antioxidant or prooxidant compounds can be formed, and interactions between food ingredients may take place. Maillard reaction products, which are formed during high-temperature heat treatment or prolonged storage, exert strong antioxidant effects [64], thus limiting MDA accumulation in ground meat [62].

The color of raw and oven-baked burgers changed rapidly in response to the tested additives and during refrigerated storage. The only exception was color lightness (L*), which remained unchanged in raw burgers and slightly decreased (significant difference) in oven-baked burgers. Surface discoloration, which is common during chilled storage [65], was not noted in raw burgers, which could be due to a short storage period (7 days). Surface discoloration was observed by Alarcón et al. [49] in deer burgers after 12 days of chilled storage. Changes in the values of color coordinates in meat products can be attributed to the oxidation of lipids and myoglobin [66,67], the growth of *Enterobacteriaceae* and *Pseudomonas* spp. [68] and a decrease in moisture retention [52]. According to Walsh and Kerry [69], meat discoloration is caused mostly by a decrease in the value of a*, and the red color determines the attractiveness of fresh meat to consumers, influencing their acceptance and governing their purchasing decisions. A decrease in the values of a* and C* in raw burgers, observed in the present study, could result from myoglobin oxidation to metmyoglobin [70] and bacterial proliferation [34] as well as the use of additives, in particular R and GR. Both raw and oven-baked burgers containing G preserved their red color (a*) better than those containing G and GR. The characteristic green color of R and the pigments contained in G affected the values of L*a*b* and, in consequence, C* and h° in burgers. The light-yellow color of fresh garlic becomes darker and more intense when its cloves are dried and powdered. Pigments of garlic cloves and dry powder contribute to color changes, and the degradation process may even produce green-yellow substances [71,72,73]. As a result, the contribution of redness (a*) decreases, and the contribution of yellowness (b*) increases [57].

### 4.2. Sensory Properties

The sensory attributes of meat products are particularly important from the consumer’s perspective and significantly influence purchasing decisions. The additives analyzed in this study had no effect on the presence of off-odors in raw burgers, which was affected only by storage time. Such relationships were also noted by Mancini et.al [8] who evaluated the development of off-odors in rabbit burgers stored for 4 and 7 days. The development of off-odors can be accelerated by the growth of microorganisms, in particular, members of the genus *Pseudomonas* [74,75] as well as *Bronchothrix thermosphacta* that impart a “cheesy” and “dairy” odor to meat, and psychrotolerant *Enterobacteriaceae* responsible for a “sulfur” odor [68]. When the above microorganisms have a high share of meat microbiota, the accumulation of their metabolites contributes to the development of off-odors [76]. In the current study, the growth of *Pseudomonas* and *Enterobacteriaceae* was observed in raw burgers*,* and off-odors were detected when bacterial counts were very low, which corroborates the findings of Mancini et al. [8]. The appearance and overall acceptability of raw burgers were modified only by the tested additives. The addition of G, R and, in particular, their combination (GR) modified the color of burgers, which affected the attributes evaluated by panelists. Similar observations were made by the cited authors [8] who analyzed the effect of garlic powder on the sensory properties of raw and cooked rabbit burgers.

Meat flavor and aroma develop during heat treatment. The compounds present in muscle tissue enter into interactions and undergo various changes such as the degradation of carbohydrates and nucleotides or lipid oxidation, leading to the development of specific meaty flavors and aromas. In the present study, the off-odors and off-flavors of oven-baked burgers changed under the influence of additives and storage time, but they were detected at very low levels. Changes in the above parameters and in aroma intensity, flavor intensity and appearance can be attributed to microbiological processes and lipid oxidation during storage. One of the main objectives of this study was to evaluate the effects of two *Allium* species on the sensory properties of rabbit meat burgers. The analyzed additives were garlic powder (G) and ramsons powder (R) because the strong flavor and aroma of fresh garlic may be unacceptable to many consumers, especially the young target audience. Garlic powder was also compared with ramsons powder because the flavor and aroma of the latter are much less powerful. The concentrations of both additives were relatively high, and they were determined based on the results of a preliminary study and a sensory analysis performed by university staff members. Moreover, such concentrations could compensate for the absence of salt in burgers; salt was excluded because it acts as a prooxidant [77]. Consumers who participated in the preliminary study and sensory panel members declared that the absence of salt in the evaluated burgers was not noticeable and had no effect on the flavor or aroma of the products. The addition of R and GR had the most beneficial influence on the sensory attributes of oven-baked burgers, including aroma intensity and flavor intensity. Rabbit burgers with the addition of 0.25% garlic powder, evaluated by Mancini et al. [8], received the lowest scores for the above sensory properties, compared with control burgers (without additives) and burgers with the addition of 1.00% salt and 0.25% garlic powder + 1.00% salt.

### 4.3. Microbiological Quality

The shelf life, nutritional value and safety of meat products are determined by the microbiological quality of meat. Muscle tissue has high water content and it is a rich source of protein, vitamins and minerals, which creates supportive conditions for microbial growth. The major sources and routes of meat contamination include the animals and their environment as well as technological operations and equipment. The level of microbial contamination of meat and meat products is affected by hygiene conditions during slaughter and carcass processing and handling, transport and storage conditions. Bacteria residing in the gastrointestinal tract and feces of livestock can contaminate carcasses during processing. Grinding also shortens the shelf life of meat because this process destroys the structure of muscle tissue and increases aeration, thus promoting the proliferation of aerobic bacteria [78].

In the present study, the tested additives (G, R and GR) had no influence on the growth of *Enterobacteriacea**,*
*Pseudomonas* spp., lactic acid bacteria or total aerobic psychrotrophic bacteria. Similar results were reported by Mancini et al. [35] who found that the addition of 0.25% garlic powder, 1.00% salt and 0.25% garlic powder + 1.00% salt to rabbit burgers had no effect on bacterial growth. However, other studies revealed the bacteriostatic effects of garlic added to meat products. Sallam et al. [58] reported that fresh garlic and garlic powder exhibited antimicrobial activities in chicken sausages stored at 3 °C—they reduced the aerobic plate count, thus extending the shelf life of the product to 21 days. Aydin et al. [79] found that fresh garlic decreased the counts of total aerobic mesophilic bacteria and coliform bacteria in ground beef refrigerated for 24 h. The results of the cited studies indicate that the bacteriostatic activity of natural additives may vary depending on their physical form, technological processes, concentrations and the type of meat to which they are added. Various natural spices and seasonings added to ground meat can exert different effects on microbial growth. Examples include the bacteriostatic activity of turmeric powder and ginger powder in rabbit burgers [9,34], yellow passion fruit co-products, tea and grape extracts in pork burgers and pork patties [80,81,82] and wine lees in deer burgers [49].

In the current study, bacterial growth in rabbit meat burgers was affected by storage time, which is consistent with the findings of other authors [35,83,84]. The counts of *Enterobacteriacea,*
*Pseudomonas* spp. and total aerobic psychrotrophic bacteria in burgers stored for 7 days were similar, and the counts of lactic acid bacteria were higher (by 0.78 log_10_ CFU/g) than those reported by Mancini et. al. [35] in rabbit burgers stored under similar conditions. Attention should be paid to the microbiological quality of the products before storage. Bacterial counts at ST0 were higher in the present experiment than in the cited studies (*Enterobacteriacea*—by 3.08 log_10_ CFU/g*,*
*Pseudomonas* spp.—by 2 log_10_ CFU/g, lactic acid bacteria—by 3.18 log_10_ CFU/g, total aerobic psychrotrophic bacteria—by 2.61 log_10_ CFU/g). This implies that the actual increase in bacterial growth during storage was lower in this study than in previous research.

Bacteria of the family *Enterobacteriaceae* are Gram-negative rods, motile and non-motile, not spore-forming; they are facultative anaerobes. An increase in the abundance of *Enterobacteriaceae* during chilled storage was also observed in raw ground pork with the addition of allspice, bay leaf, black seed, caraway, cardamom, clove or nutmeg extract [83], raw chicken meat with the addition of rosemary or clove extract [85] and raw minced pork patties with the addition of natural extracts from tea, grape, chestnut and seaweed [81].

The microbiological quality of meat is significantly affected by storage conditions. Refrigerated storage (0–4 °C) protects raw meat against excessive bacterial growth, but prolonged storage at low temperatures promotes the proliferation of psychrotrophic bacteria such as *Pseudomonas* spp. Bacteria of the genus *Pseudomonas* are Gram-negative aerobic motile rods. They colonize mainly meat surface, and can become the dominant microbiota. In the present study, the counts of *Pseudomonas* spp. and total aerobic psychrotrophic bacteria increased in stored burgers, which corroborates the findings of other authors [35,81,85,86].

Members of the genus *Lactobacillus* are Gram-positive, aerotolerant anaerobes that are a major part of the natural meat microbiota [81,85]. They ferment glucose to lactic acid. During this process, other compounds (acetic acid, ethanol, diacetyl) that enhance the flavor of meat products are also produced in low amounts. *Lactobacilli* are used in the food processing industry to produce fermented foods, but they can also contribute to food spoilage. Lactic acid bacteria produce butyric acid, succinic acid and valeric acid, which may have a negative impact on the flavor and aroma of meat. According to Mills et al. [68], total bacterial counts in meat should not exceed the threshold value of 7 log_10_ CFU per g or cm^2^. A higher microbial load can deteriorate meat quality, leading to the development of off-odors and changes in color [74]. In the current study, the above limit was not exceeded for *Enterobacteriaceae* and lactic acid bacteria, and it was slightly exceeded for total aerobic psychrotrophic bacteria and *Pseudomonas* spp. at ST7, which was also observed by Mancini et al. [35] in raw rabbit burgers stored for 7 days under similar conditions. The results of this experiment and previous studies suggest that raw rabbit meat burgers can be stored for up to 7 days. However, different storage conditions (e.g., MAP) could alter the microbiological quality of the products and extend their shelf life, but further research is needed to confirm this observation. Previous studies have investigated the microbiological quality of MAP meat products with natural plant additives, including sheep burgers [52], deer burgers [49] and beef burger patties [87], but not rabbit burgers. The effect of different types of packaging on the microbiological quality of rabbit meat has been studied [88], but MAP rabbit meat with the addition of garlic has not been analyzed to date.

## 5. Conclusions

Garlic powder and ramsons powder can be added to rabbit meat burgers to extend their shelf life and improve their eating quality. Both additives have high antioxidant potential, as confirmed by TBARS values, which remained low after storage. Garlic provided more effective antioxidant protection than ramsons, and their combination induced the highest rate of lipid peroxidation. The color of raw and oven-baked burgers was affected by the natural pigments contained in the tested additives. Changes in the color of burgers resulted also from lipid oxidation and bacterial growth during storage. Neither garlic nor ramsons exerted pronounced bacteriostatic effects when added to rabbit meat, which suggests that they should be applied at higher concentrations or in the form of extract, oil or oleoresin. Garlic powder and ramsons powder pair well with rabbit meat, and they enhanced the sensory attributes of burgers. The addition of ramsons, alone and in combination with garlic, had the most beneficial influence on the evaluated sensory properties of oven-baked burgers.

## Figures and Tables

**Table 1 animals-12-01905-t001:** pH, TBARS values (mg malondialdehyde/kg meat) and cooking loss (%) of raw and oven-baked burgers (means ± SEM).

Parameter	Additive **				Storage Time (ST) **		SEM	*p*-Value	
	C	G	R	GR	ST0	ST7		Additive	ST
raw burgers									
pH	5.79 ^a^	5.88 ^ab^	5.87 ^ab^	5.94 ^b^	5.94 ^a^	5.80 ^b^	0.012	˂0.001	˂0.001
TBARS	0.65 ^a^	0.94 ^ac^	1.07 ^c^	1.66 ^b^	0.76 ^a^	1.41 ^b^	0.058	˂0.001	˂0.001
cooking loss	15.16	17.23	15.83	16.58	16.34	16.06	0.303	0.084	0.650
oven-baked burgers									
pH	6.07 ^a^	6.11 ^a^	6.11 ^a^	6.21 ^b^	6.20 ^a^	6.05 ^b^	0.012	˂0.001	˂0.001
TBARS	1.16 ^a^	1.19 ^a^	1.57 ^b^	1.95 ^c^	1.41	1.54	0.042	˂0.001	0.125

Values within a row followed by different superscript letters, within experimental factors, are significantly different: ^a–c^—*p* ≤ 0.05. ** C—control; G—garlic (*Allium sativum* L.) powder added at 0.35 g/100 g of meat; R—ramsons (*Allium ursinum* L.) powder added at 0.35 g/100 g of meat; GR—garlic powder added at 0.35 g/100 g and ramsons powder added at 0.35 g/100 g of meat; ST—Storage time; ST0—before storage; ST7—after 7 days of storage.

**Table 2 animals-12-01905-t002:** Color parameters (L*, a*, b*, C*, h°) of raw and oven-baked burgers (means ± SEM).

Parameter	Additive **				Storage Time (ST) **		SEM	*p*-Value	
	C	G	R	GR	ST0	ST7		Additive	ST
raw burgers									
L* (lightness)	55.35 ^a^	56.77 ^b^	49.78 ^c^	49.42 ^c^	52.80	52.86	0.358	˂0.001	0.944
a* (redness)	11.50 ^a^	8.12 ^b^	2.81 ^c^	0.51 ^d^	6.87 ^a^	4.59 ^b^	0.507	˂0.001	0.023
b* (yellowness)	17.32 ^a^	18.64 ^bc^	18.19 ^b^	19.05 ^c^	18.84 ^a^	17.76 ^b^	0.109	˂0.001	˂0.001
C* (chroma saturation)	20.82 ^a^	20.34 ^a^	18.44 ^b^	19.06 ^bc^	20.55 ^a^	18.78 ^b^	0.169	˂0.001	˂0.001
h° (hue angle)	56.69 ^ab^	66.61 ^a^	81.36 ^a^	25.49 ^b^	70.91 ^a^	44.16 ^b^	5.288	0.001	0.010
oven-baked burgers									
L* (lightness)	70.06 ^a^	71.04 ^a^	66.94 ^b^	66.58 ^bc^	69.20 ^a^	68.11 ^b^	0.262	˂0.001	0.037
a* (redness)	3.98 ^a^	3.14 ^a^	0.77 ^b^	0.61 ^bc^	1.26 ^a^	2.99 ^b^	0.201	˂0.001	˂0.001
b* (yellowness)	19.79 ^a^	20.19 ^ac^	21.48 ^bc^	22.23 ^b^	19.58 ^a^	22.27 ^b^	0.221	˂0.001	˂0.001
C* (chroma saturation)	20.21 ^a^	20.44 ^a^	21.50 ^a^	13.96 ^b^	15.52 ^a^	22.53 ^b^	0.654	˂0.001	˂0.001
h° (hue angle)	78.97 ^a^	81.27 ^a^	88.04 ^a^	7.61 ^b^	45.56 ^a^	82.38 ^b^	6.129	˂0.001	0.002

Values within a row followed by different superscript letters, within experimental factors, are significantly different: ^a–c^—*p* ≤ 0.05. ** Explanation as under Table 1.

**Table 3 animals-12-01905-t003:** Sensory properties (points) of raw and oven-baked burgers (means ± SEM).

Parameter	Additive **				Storage Time (ST) **		SEM	*p*-Value	
	C	G	R	GR	ST0	ST7		Additive	ST
raw burgers									
off-odors	0.21	0.26	0.28	0.24	0.14 ^a^	0.25 ^b^	0.021	0.181	0.010
appearance	8.93 ^a^	7.72 ^b^	6.55 ^c^	5.07 ^d^	6.91	7.23	0.173	˂0.001	0.341
overall acceptability	8.44 ^a^	7.55 ^b^	7.20 ^c^	5.85 ^d^	7.36	7.16	0.112	˂0.001	0.391
oven-baked burgers									
appearance	6.35 ^a^	6.52 ^ac^	8.30 ^b^	7.50 ^bc^	6.11 ^a^	8.22 ^b^	0.158	˂0.001	˂0.001
aroma—intensity	3.72 ^a^	7.95 ^b^	8.57 ^b^	8.66 ^b^	7.12	7.33	0.258	˂0.001	0.690
flavor—intensity	5.35 ^a^	8.17 ^b^	8.37 ^b^	8.52 ^b^	7.95 ^a^	7.26 ^b^	0.161	˂0.001	0.033
hardness	5.87 ^a^	6.02 ^a^	7.70 ^b^	7.70 ^bc^	7.30 ^a^	6.35 ^b^	0.141	˂0.001	˂0.001
juiciness	5.50 ^a^	6.85 ^b^	8.05 ^c^	7.85 ^c^	7.16	6.96	0.141	˂0.001	0.484
overall acceptability	6.42 ^a^	6.90 ^a^	8.32 ^b^	7.47 ^c^	6.69 ^a^	7.60 ^b^	0.105	˂0.001	0.002
off-odors	0.15 ^ab^	0.08 ^ab^	0.07 ^a^	0.17 ^b^	0.05 ^a^	0.18 ^b^	0.013	0.040	˂0.001
off-flavors	0.42 ^a^	0.20 ^b^	0.07 ^b^	0.15 ^b^	0.12 ^a^	0.29 ^b^	0.026	˂0.001	0.001

Values within a row followed by different superscript letters, within experimental factors, are significantly different: ^a–d^—*p* ≤ 0.05. ** Explanation as under Table 1.

**Table 4 animals-12-01905-t004:** Microbiological quality (log_10_ CFU/g) of raw burgers (means ± SEM).

Parameter	Additive **				Storage Time (ST) **		SEM	*p*-Value	
	C	G	R	GR	ST0	ST7		Additive	ST
*Enterobacteriacea*	5.87	5.98	6.00	6.32	4.67 ^a^	6.42 ^b^	0.165	0.796	˂0.001
*Pseudomonas* spp.	6.04	5.64	5.85	5.65	4.23 ^a^	7.35 ^b^	0.179	0.840	˂0.001
lactic acid bacteria	5.69	5.85	6.02	5.74	5.39 ^a^	6.26 ^b^	0.072	0.388	˂0.001
total aerobic psychrotrophic bacteria	6.87	7.05	6.95	6.90	6.22 ^a^	7.66 ^b^	0.088	0.90	˂0.001

Values within a row followed by different superscript letters, within experimental factors, are significantly different: ^a,b^—*p* ≤ 0.05. ** Explanation as under Table 1.

## Data Availability

Data available on reasonable request.

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
