# Peer review of "Effects of Garlic (Allium sativum L.) and Ramsons (Allium ursinum L.) on Lipid Oxidation and the Microbiological Quality, Physicochemical Properties and Sensory Attributes of Rabbit Meat Burgers"

_animals, 2022, doi:10.3390/ani12151905_

Round 1

Reviewer 1 Report

The manuscript is presenting the effect of garlic and ramsons on lipid oxidation and the microbiological quality, physicochemical properties and sensory attributes of rabbit meat burgers. The manuscript is very interesting and fits in the scope of the Animals.

The individual chapters provide information on rabbit quality meat and the role of natural antioxidant substances added to food, allowing for the extension of the shelf life in refrigerated conditions.

In the reviewer’s opinion the objective of the study is correctly formulated and sufficiently described. The experimental design and methods are appropriate for the purposes of the study, and investigations has been conducted in an ethically acceptable manner. The abstract of the reviewed paper has informative and fully covered the content of the manuscript. The length of the manuscript is appropriate in relation to the content.

The article has been prepared in accordance with the instructions for authors. English is understandable. This article requires some minor clarifications prior publication in Animals. After reading through the manuscript, I found several issues that should be addressed by the authors:

Reviewer Comments to Authors:

Line 134 – What was the average life weight of the animals?

Line 135-137 – What was the caloric value of the feed, expressed in megajoules (MJ) of ME per 1.0 kg?

Line 140 – “Carcasses ….” rather ….. “Slaughtered rabbits/animals”.

Line 146-147 – What material (-s) are the bags made of? How was vacuum pressure value in bag (expressed in % and kPa)? What was the permeability of the packaging film in relation to O2 and moisture vapor (expressed in x cm3´m2´24h´x MPa)?

Line 148 – What method was used to freeze the sample? What was the freezing time?

Line 150 – What method was used to thaw the samples? What was the thawing time?

Line 186 – “Cooking loss” ….. I suggest changing on “Thermal loss”.

Line 187 – “After cooking ……” please changing on “After roasting …”.

After a careful review of the manuscript's scientific merit, I believe that the paper is suitable for publication after correction according to reviewer recommendation.

Author Response

Dear Reviewer,

Thank you very much for the kind review of our article. We provide answers to your questions and comments below.

Yours sincerely,

Authors

Line 134 – What was the average life weight of the animals?

The body weight was 2503,77±22.48g (means ± SEM). The weight of a cold carcass without a head and giblets was 1198.15±31.86g (means ± SEM)

Line 135-137 – What was the caloric value of the feed, expressed in megajoules (MJ) of ME per 1.0 kg?

Metabolic energy (ME) of feed was 10.2 MJ.

Line 140 – “Carcasses ….” rather ….. “Slaughtered rabbits/animals”.

Of course, there should be "Slaughtered rabbits". "Carcasses" was used by mistake.

Line 146-147 – What material (-s) are the bags made of? How was vacuum pressure value in bag (expressed in % and kPa)? What was the permeability of the packaging film in relation to O2 and moisture vapor (expressed in x cm3´m2´24h´x MPa)?

Meat samples were packaged in vacuum (99% vacuum is equal to 1.3 KPa) in bags made of polyamide/polyethylene film (permeability: O2 = 25 cm3/m2/24 h/0.1 Mpa; CO2 = 85 cm3/m2/24 h/0.1 Mpa; N2 = 7 cm3/m2/24 h/0.1 Mpa; moisture vapor <3 g/m2/24 h).

Line 148 – What method was used to freeze the sample? What was the freezing time?

Samples were frozen at –22°C (slow freezing, 0.2–1 cm/h) for 20 h.

Line 150 – What method was used to thaw the samples? What was the thawing time?

The samples were thawed in the atmospheric air. The thawing time (20 h) was given in the manuscript (line 153).

Line 186 – “Cooking loss” ….. I suggest changing on “Thermal loss”.

However, we would prefer to stay with "cooking loss" because it is a typical term describing weight losses during thermal treatment, while "thermal" is used, for example, for such terms as: thermal conductivity of meat, thermal energy, thermal treatmet.

Line 187 – “After cooking ……” please changing on “After roasting …”.

Replaced as suggested by the reviewer.

Reviewer 2 Report

animals-1822116

Manuscript review:

Title: Effects of garlic (Allium sativum L.) and ramsons (Allium ursinum L.) on lipid oxidation and the microbiological quality, physicochemical properties and sensory attributes of rabbit meat burgers.

General comments:

The authors presented research on evaluation of rabbit meat quality and its properties when seasoned with garlic and ramsons powders. Food additives were used individually or in combination. Several meat quality parameters were measured; therefore the research is very comprehensive and complete.

The less known species of the Allium genus has been tested in comparison to common garlic, which is used worldwide in traditional and modern cuisine for its wide range of beneficial properties. Ramsons is gaining more attention among consumers, therefore studying it is well justified. Although all procedures and research methodology are very typical for the evaluation of meet for human consumption, the reviewed manuscript represents a unique research on the rabbit meat and a novel food additive.

Manuscript is generally well written. Design of the study is straightforward and statistical procedures are used correctly. The methodology is wide-ranging and well described. The complexity of the evaluation is impressive, and research was well conducted. Results are properly presented and discussed. Appropriate number, quality and relevance of referenced literature was used.

I don’t have any specific comments regarding the manuscript. Well done!

Author Response

Dear Reviewer,

Thank you very much for the kind review of our article.

Yours sincerely,

Authors